# Effects of Coumarin on Rhizosphere Microbiome and Metabolome of *Lolium multiflorum*

**DOI:** 10.3390/plants12051096

**Published:** 2023-03-01

**Authors:** Yihu Yang, Jun Xu, Yan Li, Yuchen He, Yuqing Yang, Dalin Liu, Caixia Wu

**Affiliations:** College of Animal Science and Technology, Yangzhou University, Yangzhou 225009, China

**Keywords:** coumarin, annual ryegrass, physiological characteristics, rhizosphere microecology

## Abstract

Rhizosphere microorganisms can help plants absorb nutrients, coordinate their growth, and improve their environmental adaptability. Coumarin can act as a signaling molecule that regulates the interaction between commensals, pathogens, and plants. In this study, we elucidate the effect of coumarin on plant root microorganisms. To provide a theoretical basis for the development of coumarin-derived compounds as biological pesticides, we determined the effect of coumarin on the root secondary metabolism and rhizosphere microbial community of annual ryegrass (*Lolium multiflorum* Lam.). We observed that a 200 mg/kg coumarin treatment had a negligible effect on the rhizosphere soil bacterial species of the annual ryegrass rhizosphere, though it exhibited a significant effect on the abundance of bacteria in the rhizospheric microbial community. Under coumarin-induced allelopathic stress, annual ryegrass can stimulate the colonization of beneficial flora in the root rhizosphere; however, certain pathogenic bacteria, such as *Aquicella* species, also multiply in large numbers in such conditions, which may be one of the main reasons for a sharp decline in the annual ryegrass biomass production. Further, metabolomics analysis revealed that the 200 mg/kg coumarin treatment triggered the accumulation of a total of 351 metabolites, of which 284 were found to be significantly upregulated, while 67 metabolites were significantly downregulated in the T200 group (treated with 200 mg/kg coumarin) compared to the CK group (control group) (*p <* 0.05). Further, the differentially expressed metabolites were primarily associated with 20 metabolic pathways, including phenylpropanoid biosynthesis, flavonoid biosynthesis, glutathione metabolism, etc. We found significant alterations in the phenylpropanoid biosynthesis and purine metabolism pathways (*p <* 0.05). In addition, there were significant differences between the rhizosphere soil bacterial community and root metabolites. Furthermore, changes in the bacterial abundance disrupted the balance of the rhizosphere micro-ecosystem and indirectly regulated the level of root metabolites. The current study paves the way towards comprehensively understanding the specific relationship between the root metabolite levels and the abundance of the rhizosphere microbial community.

## 1. Introduction

Coumarins are primarily plant-derived secondary metabolites with allelopathic properties and have been proven to exert a phytotoxic effect. Coumarins are produced by many plant species, such as species belonging to *Umbelliferae*, *Rutaceae*, *Leguminosae*, and *Compositae* [1,2,3]. These aromatic compounds can be found in different plant parts, including the roots, stems, leaves, flowers, seeds, and fruits of plants [4]. Among these, the coumarin content is highest in flowers and fruits. It is also distributed in small amounts in root exudates [5]. Previous studies have shown that different coumarins have inhibitory effects on plant growth at varying concentrations. Razavi and Zarrini [6] revealed that 100 μg/mL of furanocoumarin can inhibit seed germination, seedling growth, and root crown elongation in lettuce (*Lactuca sativa* L.). Baskin et al. [7] found that psoralen extracted from *Psoralea corylifolia* L. can inhibit the germination and root growth of its seeds as well as other plant seeds. Coumarin was also found to reduce the germination rate, water absorption rate, electrolyte retention capacity, and level of oxygen consumption in wheat (*Triticum aestivum* L.) seeds [8]. In our earlier study, we showed that *Melilotus officinalis* has a very strong inhibitory effect on Italian ryegrass (*Lolium multiflorum* Lam.), Arab speedwell (*Veronica persica* Poir.), annual bluegrass (*Poa pratensis* L.), perennial weed dandelion (*Taraxacum officinale* L.), common sow thistle (*Sonchus oleraceus* L.), broomsedge (*Kochia scoparia* L. Schrad.), flixweed (*Descurainia sophia* L. Webb. ex Prantl), prickly Russian thistle (*Salsola tragus* L.), and cheatgrass *(Bromus tectorum* L.) [9]. The isolation and identification of aqueous extract from *M. officinalis* revealed that coumarin (2H-1-benzopyran-2-one) was the predominantly present allelochemical in the extract and that coumarin is the cause of growth inhibition [10].

A plethora of studies have shown that coumarin also has allelopathic effects on the growth of plant roots [11,12,13]. Roots are important organs that enable plants to absorb water and mineral nutrients, and coumarins are plant-derived chemicals that can cause changes in the edaphic properties. The effects of coumarins on roots vary based on their concentration in and the root type of plants. Abenavoli et al. [11] found that 400 μM of coumarin inhibited root growth in corn seedlings. In a subsequent study, Abenavoli et al. [12] evaluated the toxic effects of trans-cinnamic acid, umbelliferone, and coumarin on *Arabidopsis thaliana* roots using a non-linear regression analysis. The findings revealed that a low concentration of coumarin promoted the elongation of lateral roots, which is consistent with a previous study where the allelopathic effect of coumarin was found to be dependent on the root type. Furthermore, Abenavoli et al. [13] also reported that coumarin caused swelling in the lower surface of the root tip exclusively at a concentration of 0.1 mM, which led to an increased surface area of the root and enhanced the ability of the root to absorb nitrate.

Coumarin also serves as a signaling molecule that regulates the interaction between symbiotic or pathogenic bacteria and plants. Several species of pathogenic bacteria can induce the accumulation of coumarin-derived compounds around plant roots and stems. These compounds can be used as protective agents that immunize the plant against the invasion and propagation of pathogens [14]. Studies have shown that various plants can synthesize coumarin compounds in vivo as a result of pathogen infection [15,16]. Coumarin compounds have shown inhibitory activities against several plant pathogens. Scopolamine exhibited strong inhibitory activities against *Ophiostoma ulmi spores*, *Cercospora nicotianae*, *Botrytis cinerea*, *Alternaria alternata*, *Phytophthora parasitica* var. *nicotianae*, and *Pseudomonas syringae* [17,18,19]. In addition, furanocoumarins, psoralen, and bergapten also showed antibacterial activities [20]. Kwon et al. [21] confirmed that seven furanocoumarins produced by *Angelica dahurica* exerted significant antibacterial effects on *Escherichia coli*, *Bacillus subtilis,* and *Clodosporium herbarum*. Integrated metabonomics and metagenomics analyses of Arabidopsis roots illustrated that the root-specific transcription factor MYB72 can promote the production of coumarin scopolamine and, consequently, its antibacterial activity in root exudates. Coumarins can also affect the microbial community structure around plant roots and suppress the growth of fungal pathogens in soil. Al-Barwani and Eltayeb [22] compared the antibacterial properties of xanthotoxin, sweet internal lipid, 7-hydroxycoumarin, scopolamine, and angelicin, and revealed that these compounds had certain inhibitory effects on *Bipolaris* spp., *Fusarium* spp., and *Alternaria* spp. Coumarin compounds can selectively affect the colonization of plant root flora communities around the roots and inhibit the growth of soil-borne fungal pathogens such as *F.oxysporum f.* sp. *raphani* and *Verticillium dahliae* [23].

Annual ryegrass (*Lolium multiflorum* Lam.), an annual or interannual herb, is a malignant weed that is grown in winter crop fields. In the past few years, many advances have been made in the research and development of new herbicides that can be used to manage annual ryegrass [24]. Yao et al. [25] showed that 100 μg/mL of coumarin can significantly inhibit the germination of *Lolium multiflorum* L. seeds and the early growth of seedlings. It can destroy the membrane structure of seed endosperm cells, and inhibit their decomposition and nutrient supply, causing these cells to lose water and the separation of their cell walls. Moreover, coumarin can influence the growth of plant roots easily through an allelochemical inhibitory effect. Therefore, analyzing the composition and quantity of root exudates could provide insight into the chemical and biological processes occurring in the rhizosphere. Root exudates are majorly composed of primary and secondary metabolites. Of these, secondary metabolites can affect the biodiversity of rhizosphere microorganisms [23,26]. In the current study, we speculated that coumarin affects the microbial diversity in the rhizosphere and the composition of metabolites in annual ryegrass roots. Our aim was to investigate the effects of coumarin on the microbial diversity in the soil and the metabolite composition of perennial ryegrass roots.

## 2. Results

### 2.1. Diversity in the Bacterial Community of the Rhizosphere

We analyzed the composition and diversity of the bacterial community in the rhizosphere and found that (Appendix A) a total of 6511 OTUs were detected in the CK group, while about 6568 OTUs were detected in the T200 group. Of these, about 5658 OTUs were commonly detected between both groups. Though there was no significant difference observed between the CK and T200 groups in terms of the Shannon and Simpson indices (*p* > 0.05), the richness and chao1 and ACE values in the T200 group were markedly higher than those in the CK group (*p <* 0.05) (Table 1). Further, anosim and Unweighted UniFrac distance analyses (*p =* 4.303) indicated no significant difference between the bacterial communities of the CK and T200 groups (Figure 1). It was apparent that coumarin had a negligible effect on the diversity of the rhizosphere bacterial community of annual ryegrass.

### 2.2. Comparison of Relative Abundance of Bacterial Communities in Rhizosphere Soil under Phylum and Genus Classification

We observed significant differences in the relative bacterial abundance at the phylum level between the T200 and CK groups (*p <* 0.05) (Appendix A). Among the phyla identified in the samples, the relative abundance levels of Proteobacteria (*p* = 0.048), Planctomycetes (*p* = 0.028), Nitrospirae (*p* = 0.029), and Elusimicrobia (*p* = 0.017) were significantly higher in the T200 group compared to the CK group. Figure 2 depicts that the relative abundance levels of unidentified*_Nitrospiraceae*, *Polycyclovorans*, *Stenotrophobacter*, *Cupriavidus*, *Chthoniobacter*, unidentified*_Alphaproteobacteria, Ramlibacter*, *Geobacter*, *Marmoricola*, *Bdellovibrio*, unidentified*_Bactreia*, *Azoarcus*, *Aquicella*, and *Blastococcus* were notably higher in the T200 group than those in the CK group (*p <* 0.05). In contrast, the relative abundance of *Massilia* and *Neorhizobium* was lower in the T200 group compared to the CK group (*p <* 0.05).

### 2.3. Non-Targeted Metabolomics Analysis of Roots Using LC-MS

PCA and PLS-DA analyses showed that the root metabolites of the CK and T200 groups were significantly different, as shown in Figure 3a,b. A total of 1477 metabolic compounds were detected in the root samples, and the KEGG pathways were annotated for about 181 metabolites (Appendix A). The differential expression of a large number of metabolites in the T200 and CK groups can also be distinguished in the volcanic diagram and heat map clustering (Figure 3c,d). Based on the t-test, about 284 metabolic compounds had significantly higher content in the T200 group than in the CK group, while 67 metabolites were decreased in the T200 group compared to those in the CK group (*p* < 0.05).

### 2.4. Differential Metabolite Analysis in Roots through Non-Targeted Metabolomics

The differential metabolites identified in the current study are given in Table 2. The content of cinnamic acid, tryptophol, codeinone, 7-Methoxy-4-methylcoumarin, lactide, allantoic acid, coumarin, xanthurenic acid, eriocitrin, salidroside, bis (glutathionyl) spermine, and bis (glutathionyl) spermine disulfide was found to be significantly increased in the T200 group compared to that in the CK group (*p <* 0.05), whereas the levels of metabolites, such as quercetin, 7-ACA, chlorogenic acid, and glycerophosphoglycerol, were upregulated in the CK group compared to the T200 group (*p <* 0.01).

### 2.5. KEGG Pathway Enrichment Analysis of Differentially Expressed Metabolites in Roots

The differential metabolites identified between CK and T200 groups were found to be associated with the pentose phosphate pathway, carbon metabolism, anthocyanin biosynthesis, acridone alkaloid biosynthesis, tyrosine metabolism, tryptophan metabolism, isoquinoline alkaloid biosynthesis, glutathione metabolism, flavonoid biosynthesis, phenylpropanoid biosynthesis, and secondary metabolite biosynthesis (Figure 4). In addition, Phenylpropane biosynthesis and the purine metabolism pathways were significantly enriched (*p <* 0.05).

### 2.6. Correlation Analysis between Bacterial Genera in the Rhizosphere and Root Metabolites

Correlations between the bacterial genera colonized in the rhizosphere soil and root metabolites in the T200 and CK groups were analyzed in the current study, and the Figure 5 showed that *Aquicella*, *Bdellovibrio*, *Geobacter*, *Candidatus_Omnitrophus*, *Bacteriovorax*, *Polyangium*, *Azoarcus*, *Desulfuromonas*, unidentified*_Bacteria,* and *Blastocatella* were positively correlated with 1,2,4-Benzenetriace, Benzaldehyde, Cinnamic acid, Benzoyl-beta-D-glucoside, Setileuton, isopropylmalic acid, (2E)-2-(4-Chlorophenyl)-3-phenyl-3-(2-pyridinyl)acrylonitrile, Acetophenone, Dexloxiglumide, Salicin 6-phosphate, Embramine, 7-Methoxy-4-methylcoumarin, Mps1-IN-1, dimethyl itaconate,(2R)-1-[(2R)-2-Piperidinyl]-2-pentanol, and Coumarin (*p <* 0.05), while these bacteria were negatively correlated with N-Hydroxydecanamide (*p <* 0.05).

## 3. Discussion

The rhizosphere is the shallow zone in the soil surrounding the plant roots that is directly affected by root exudates. The rhizosphere is one of the most complex ecosystems, comprising plant roots, soil, and diverse microbes. It has an important influence on the growth and development of plants and their immunity [27]. Several studies have shown that rhizosphere microbiota can enhance the fluidity of soil nutrients such as nitrogen, phosphorus, and potassium, thereby promoting the absorption of nutrients by plant roots and influencing plant growth [28,29,30]. Rhizosphere microbes also serve as an important line of defense for plants against pathogens. Previous studies have shown that microbes residing in the rhizosphere can aid the detoxification of plants and their resistance to the attack of soil-borne pathogens, as well as enhance their environmental adaptability [31,32]. Thus, a complex interaction has been established between plants and rhizosphere microbiota. These microorganisms, as symbionts, play an important role in plant growth and stress resistance. The microbial composition in rhizosphere soil can also reflect the health of the soil [33,34]. During normal plant growth, the composition and proportion of microbial communities in the rhizosphere are dynamically balanced. Under stress conditions, this balance is perturbed, which leads to changes in the composition of rhizosphere bacterial communities [35]. Niro et al. [36] showed that coumarin did not influence the Shannon and Simpson indexes, while the observed species, chao1, and ACE indexes were promoted by the coumarin treatment, which suggested that coumarin may affect the richness rather than the diversity of microbial communities in rhizosphere soil. In the current study, no significant differences were observed in the species or Shannon indexes between the T200 and CK groups, indicating that coumarin had no significant effect on the richness and diversity of the bacterial community in rhizosphere soil,

At the phylum level, *Proteobacteria*, *Actinobacteria*, *Planctomycetes*, and *Firmicutes* were the dominant flora identified in the rhizosphere soil samples of both the CK and T200 groups, which is consistent with the findings of Cheng et al. [37] and Wang et al. [38]. Further, our results showed that the relative abundance of Proteobacteria in the T200 group was higher than that in the CK group. Pseudomonadaceae is a kind of Gram-negative bacteria widely distributed in the soil and is one of the common bacterial groups in the rhizosphere of plants. Studies have shown that Pseudomonadaceae can produce effective iron carriers, antibiotics, and other metabolites, which can resist the damage induced by pathogenic microorganisms towards plants [39]. At present, most of the research on Pseudomonadaceae focuses on the bacteriostatic activity of plant-pathogenic fungi. There are few reports on the regulation of plant growth and development under adverse stress. We speculate that this kind of bacteria starts an immune response mechanism and propagates in large numbers under allelopathic stress, and that it interacts with other microorganisms in the rhizosphere soil to resist stress damage and promote plant growth. Its specific response mechanism requires further study. Many of these bacteria have a nitrogen fixation ability and can produce polycyclic aromatic hydrocarbons that are beneficial for plant growth and development [40]. Therefore, the proliferation of Proteobacteria under allelopathic stress induces stress resistance in annual ryegrass. Unlike the CK group, the relative abundance of the Pseudomonadaceae family was significantly increased from 0.0143 to 0.0832 in the T200 group (Appendix A). Nagel et al. [41] demonstrated that the bacteria related to the Pseudomonadaceae family can prevent and control plant diseases and promote plant growth. At the genus level, the relative abundance of *Geobacter*, *Aquicella,* and unidentified*_Nitrospiraceae* was significantly increased in the T200 group compared to the CK group. *Geobacter* is an important iron-reducing bacteria. Numerous studies have shown that coumarin compounds and their derivatives can decrease the impact of environmental stress in plants by chelating iron or promoting the uptake of iron by roots [42,43,44]. Thus, it can be speculated that coumarin can improve plants’ iron absorption capacity by promoting the accumulation of *Geobacter* species. Since *Nitrospiraceae* is known to fix atmospheric nitrogen, unidentified*_Nitrospiraceae*, detected in our study, can also improve the tolerance of annual ryegrass to allelopathic stress. In contrast, *Aquicella*, a unique pathogenic bacterium, has a negative impact on plant growth and development [45]. Under allelopathic stress, *Aquicella* species propagate in large numbers. From these observations, we speculated that one of the possible reasons for this phenomenon could be that stress exposure leads to alterations in the rhizosphere soil microbial community, which has an inhibitory effect on the reproduction of certain microbial communities. These microorganisms have an antagonistic effect on the growth of *Aquicella*, which leads to the mass propagation of *Aquicella* species. In addition, plant roots secrete a few substances under allelopathic stress that are selective for *Aquicella*, such as organic acids, amino acids, and phenols, in order to promote its mass reproduction [46]. Altogether, the accumulation of *Aquicella* species and the secretion of these specific compounds could be an important reason behind a sharp decline or even death of annual ryegrass biomass.

In the current study, 351 metabolites were found to be significantly different between the T200 and CK groups. Further, the KEGG enrichment analysis of the metabolic pathways revealed that the metabolites were involved in the biosynthesis of phenylpropane, as well as in metabolic pathways such as the Pentose phosphate pathway, carbon metabolism, and tryptophan metabolism. The Pentose phosphate pathway not only mediates the oxidative decomposition of glucose but also serves as an important multifunctional metabolic pathway that provides NADPH to maintain plant growth and promote other biosynthetic and metabolic processes [47]. Few studies have demonstrated that during environmental stress, the pentose phosphate pathway can enhance metabolic processes in plants. de Freitas-Silva et al. [48] revealed that the oxidative stress induced by glyphosate can trigger the pentose phosphate pathway in *Arabidopsis thaliana.* It was speculated from the observations that in the pentose phosphate pathway, the oxidation phase of NADPH (OxPPP) maintained the activity of glutathione reductase (GR), thereby inducing the preservation and regeneration of GSH in the cells under glyphosate-induced stress. These results were consistent with our previous findings that coumarin has an inhibitory effect on the growth (root length, biomass, and MDA content) of annual ryegrass (*Lolium multiflorum* Lam.), which is directly proportional to the concentration of coumarin, thus explaining the effect of coumarin on the glutathione metabolism pathway in roots [49]. Moreover, the pentose phosphate pathway is also known to play an important role in the post-embryonic development of *Arabidopsis thaliana* [50]. In the current study, the metabolic compounds that showed differential expression were mainly associated with tryptophan metabolism (Figure 4). Tryptophan is an important synthetic precursor of plant growth hormone (IAA), melatonin, and serotonin; therefore, it can induce and regulate various phytochemical and morphological pathways, thus playing a crucial role in plant growth [51]. El Karamany et al. [52] showed that different concentrations of tryptophan (50, 75, and 100 mg/L) improved the yield and nutritional value of Berseem green plants. Therefore, coumarin can affect the energy conversion and auxin synthesis of ryegrass by regulating the phosphate pathway, carbon metabolism, tryptophan metabolism, and other metabolic pathways, thus affecting plant growth and metabolism.

Environmental stress leads to the induction of oxidative stress response in plants, which further affects plant growth and metabolism [48]. However, flavonoids are essential compounds that are capable of scavenging free radicals and antioxidants. Among all flavonoids, anthocyanin is the most effective antioxidant and free radical scavenger known in plants [53]. Flavonoids can also promote the production of glutathione (GSH), superoxide dismutase (SOD), catalase (CAT), and other antioxidant enzymes to eliminate excessive free radicals and inhibit the peroxide process in plants [54]. Our findings revealed that coumarin can influence the anthocyanin synthesis, GSH metabolism, and flavonoid biosynthesis pathways in ryegrass. In addition, the accumulation of coumarin inevitably stimulated oxidative stress in the ryegrass roots.

Benzene metabolism plays a vital role in the physiological activities of enzymes and intermediate products. Compounds such as lignin and phenol provide plants with protection by regulating cell differentiation, inhibiting pathogen infection, and promoting pigment formation [53]. Both phenylalanine and phenylpropionic acid CoA esters involved in the phenylpropane metabolic pathway can produce alkaloid compounds. In addition, phenylpropionic acid CoA and malonamide CoA, which are produced during acetic acid metabolism, can synthesize flavonoids and isoflavones, while ρ-coumaric acid, which is produced by phenylalanine metabolism, can be converted into coumarin [55]. Johnson and Schaal [56] identified chlorogenic acid in a potato tuber resistant to scab as early as 1952. Further, Uritani and Miyano [57] isolated chlorogenic acid from the healthy tissue of sweet potatoes infected with black spot fungus. In addition, Johnson and Schaal [58] also detected the accumulation of chlorogenic acid in mechanically damaged potato tubers. It was shown that chlorogenic acid has antibacterial activity against both bacteria and fungi [59]. These confirmed that chlorogenic acid is closely related to plant stress resistance. This study found that the content of chlorogenic acid present during root metabolism was significantly down-regulated, indicating that coumarin allelopathic stress caused an abnormality in the phenylpropanoid metabolic pathway and inhibited the biosynthesis of chlorogenic acid. In addition, the content of coumarin in the root metabolites was significantly increased. A large number of studies [60] have shown that high concentration of coumarin can damage the root antioxidant system, and that the accumulation of reactive oxygen species in root cells can lead to an increase in cell membrane permeability and the destruction of plant tissue, thus inhibiting the development of roots.

In the current study, the correlation analysis between rhizosphere bacterial community and root metabolites revealed that *Aquicella*, *Bdellovibrio*, *Geobacter,* and ten other genera of bacteria were positively correlated with 1,2,4-Benzenetriace, benzaldehyde, cinnamic acid, and 18 metabolites; however, no positive correlation was observed with N-Hydroxydecanamide. The above metabolites are mainly concentrated in plant-resistant substances, such as N-hydroxydecanamide, coumarin, and 7-methoxy-4-methylcoumarin, and plant growth regulators, such as cinnamic acid, N-hydroxyphenylacetylglutamic acid, and benzoyl β-D-glucoside. The rhizosphere is the center of interaction between the plant roots and soil microorganisms and is directly influenced by the physiological and biochemical activities of plant roots. It regulates the compounds involved in these activities. In the current study, we illustrated that under coumarin-induced allelopathic stress, the abundance and composition of soil microbes are altered, which disrupts the balance of the rhizosphere micro-ecosystem, indirectly regulating the concentrations of root metabolites. However, a comprehensive investigation is required to understand how the rhizosphere microbiota regulates root metabolism.

## 4. Conclusions

In this study, under coumarin-induced allelopathic stress, the relative abundance of rhizosphere-beneficial bacteria was found to be stimulated in order to resist the stress conditions in annual ryegrass. However, the accumulation of a few pathogenic bacteria, such as *Aquicella* species, was also observed under coumarin-induced stress. The metabolome analysis unraveled a total of 284 metabolites that were more significantly upregulated in the T200 group compared to the CK group, while only 67 compounds were downregulated to a lower degree in the T200 group compared to the CK group. These metabolites were enriched with respect to the Pentose phosphate pathway, carbon metabolism, anthocyanin biosynthesis, glutathione metabolism, and flavonoid biosynthesis, which are mainly related to energy metabolism and antioxidant activity in annual ryegrass. Further, some rhizosphere soil bacteria exhibited significant correlations with several root metabolites, although the specific mechanism behind these correlations needs further investigation.

## 5. Materials and Methods

### 5.1. Materials and Plant Treatments

Coumarin was purchased from SIGMA (purity ≥ 99%, NO.C4261-50G). The seeds of annual ryegrass (*Lolium multiflorum* Lam.) were procured from the Grass Science Laboratory of Yangzhou University. The potting soil was collected from surface cultivation soil (0~20 cm) from the pilot fields of Yangzhou University. The soil texture was sandy loam, and its pH was 7.07. Plastic pots with an upper diameter of 16 cm, a lower diameter of 12 cm, and a height of 12 cm were used to grow the plants.

Plants treated with a coumarin concentration of 200 mg/kg (coumarin was introduced in the soil under a simulated natural environment, and 200 mg of coumarin was used to treat each kg of soil) and a distilled water treatment (the control treatment) were recorded as T200 and CK groups, respectively, and each treatment was repeated 7 times. A total of 40 annual ryegrass seeds were selected, and after disinfection by HgCl_2_, seeds were evenly sown in a flowerpot containing 1.6 kg of dried soil. The flowerpot was randomly moved every other day to ensure uniform light and water management. The seedlings were screened after two weeks of growth, and 30 plants with the same growth vigor were retained in each pot for coumarin treatment. Samples were collected after 20 days of the coumarin treatment.

### 5.2. Sampling of Roots and Rhizosphere Soil of Annual Ryegrass

Rhizosphere soil samples were collected through the shaking method, after removing the non-rhizosphere soil, and then packed in a 15 mL centrifuge tube, frozen in liquid nitrogen, and stored at −80 °C for subsequent experiments. After the collection of rhizosphere soil samples, the roots were rinsed with distilled water and cut with scissors. The roots of each group were divided into 6 parts, wrapped in foil paper, marked, and then frozen in liquid nitrogen quickly and stored at −80 °C for metabolome analysis.

### 5.3. DNA Extraction and Sequencing

The SDS method was used to extract the total DNA of rhizosphere soil samples. The samples were ground to break their tissues and cells, and then SDS was used to split them. Then, a high concentration of potassium acetate was added, and the samples were stored at 0 °C to remove protein and polysaccharide impurities; finally, ethanol or isopropanol was used to precipitate the DNA. The treated samples were then stored on standby. The purity levels of the DNA were determined by agarose gel electrophoresis. An appropriate amount of the sample DNA was extracted and placed in a centrifuge tube, and the samples were diluted to 1 ng/μL with sterile water. PCR amplification was performed using Phusion^®^High-Fidelity PCR Master Mix (New England Biolabs, Ipswich, MA, USA) according to the manufacturer’s instructions. The diluted total DNA was used as a template for PCR amplification, and the primer used in this study was 515F-806R. The purification of the PCR product was performed using a GeneJET gel kit (Thermo Fisher Scientific, Waltham, MA, USA). The Ion Plus Fragment Library Kit 48 reactions (Thermo Fisher Scientific, Waltham, MA, USA) was used to construct the library, and sequencing was performed on an Ion S5TMXL platform.

### 5.4. Sequencing Data Analysis

Cutadapt (V1.9.1) [61] was used to remove low-quality reads, and the barcode was used to obtain sample data. Further, the trimming of the barcode and an analysis of preliminary quality were used to obtain the raw reads. The annotation database (LIPID MAPS, https://www.lipidmaps.org/ (accessed on 20 June 2022)) was searched to detect the chimeric sequences in the microbial species through Vsearch [62], and chimeric sequences were removed to obtain the clean reads, as described by Haas et al. [63]. UPARSE software (V7.0.1001) was used to cluster all clean reads with 97% consistency to obtain OTU sequence clustering (the Operational Taxonomic Units) [64]. Mothur and SILVA132 SSUrRNA databases were used for species annotation analysis (threshold 0.8~1) [65,66], and MUSCLE software (V 3.8.31) was used for multiple sequence alignment [67]. Finally, the data of each sample were rarified to the sample with lowest read number. The conducted α-diversity and β-diversity analyses were based on these data. We used the R package (v3.6.1) to perform the α-diversity index, β-diversity, and difference (Wilcox.test) analyses of the dataset. Further, we used the analysis of similarities (Anosim) method and unweighted UniFrac distance method to interpret the β-diversity. Significant differences between both groups were analyzed by Wilcox.test in STAMP (V2.1.3), and *p <* 0.05 was considered significant.

### 5.5. Root Metabolites Extraction and LC-MS Analysis

Root tissues (100 mg) were individually ground with liquid nitrogen, and the homogenate was re-suspended in pre-chilled 80% methanol and 0.1% formic acid by vortexing. The samples were incubated on ice for 5 min and then centrifuged at 15,000 rpm and 4 °C for 5 min. The supernatant was diluted to a final concentration containing 60% methanol via LC-MS grade water. The samples were subsequently transferred to a fresh Eppendorf tube after filtration with a 0.22 μm filter and then centrifuged at 15,000× *g* and 4 °C for 10 min. Finally, the filtrate was injected into the LC-MS/MS system.

LC-MS analysis was performed using a Vanquish UHPLC system (Thermo Fisher) coupled with an Orbitrap Q Exactive series of the mass spectrometer (Thermo Fisher, Waltham, MA, USA). Samples were injected in a Hyperil Gold column (100 × 2.1 mm, 1.9 μm) using a 16 min linear gradient at a flow rate of 0.2 mL/min. The eluents for the positive polarity mode were eluent A (0.1% FA in Water) and eluent B (Methanol). The eluents for the negative polarity mode were eluent A (5 mM ammonium acetate, pH 9.0) and eluent B (Methanol). The solvent gradient was set as follows: 2% B, 1.5 min; 2–100% B, 12.0 min; 100% B, 14.0 min; 100–2% B, 14.1 min; and 2% B, 16 min. Q Exactive mass spectrometer was operated in positive/negative polarity mode with a spray voltage of 3.2 kV, capillary temperature of 320 °C, sheath gas flow rate of 35 arb, and aux gas flow rate of 10 arb.

### 5.6. Metabolomics Data Analysis

The raw data files generated by UHPLC-MS were processed using Compound Discoverer 3.0 (CD3.0, Thermo Fisher, Waltham, MA, USA) to perform peak alignment, peak picking, and quantitation for each metabolite. The main parameters were set as follows: retention time tolerance, 0.2 min; actual mass tolerance, 5 ppm; signal intensity tolerance, 30%; signal/noise ratio, 3; and minimum intensity, 100,000. After processing, peak intensities were normalized to the total spectral intensity. The normalized data were used to predict the molecular formulae based on additive ions, molecular ion peaks, and fragment ions. Then, peaks were matched with the mzCloud (https://www.mzcloud.org/ (accessed on 14 May 2022)) and ChemSpider (http://www.chemspider.com/ (accessed on 14 May 2022)) databases to obtain accurate qualitative and relative quantitative results.

Metabolites were annotated using the KEGG database (http://www.genome.jp/kegg/ (accessedon 14 May 2022)). Principal components analysis (PCA) and Partial least squares discriminant analysis (PLS-DA) were performed using metaX [68,69]. We used a t-test to calculate the statistical significance of the dataset. The metabolites with *p*-values < 0.05 were considered to be differentially expressed metabolites. Volcano plots were used to filter metabolites of interest, which were based on Log_2_(FC) and −log^10^(*p*-value) values of metabolites. For heat map clustering, the data were normalized using *z*-scores of the intensity areas of differential metabolites and plotted using the Pheatmap package in R language. The functions of these metabolites and metabolic pathways were analyzed using the KEGG database [70,71]. Further, metabolic pathway enrichment analysis of differential metabolites was performed; when the ratio was satisfied by x/n > y/N, a metabolic pathway was considered to be enriched; when the P-value of the metabolic pathway was <0.05, a metabolic pathway was considered to be statistically significant. Correlation analysis was performed between the significantly different flora obtained by 16S rDNA analysis and metabolites obtained by metabolomics analysis based on the Pearson correlation coefficient to measure the degree of correlation between species diversity and metabolites in the rhizospheric samples. Further, the R program (3.6.1) was used to complete the correlation analysis between bacterial communities and metabolomes.

## Figures and Tables

**Figure 1 plants-12-01096-f001:**
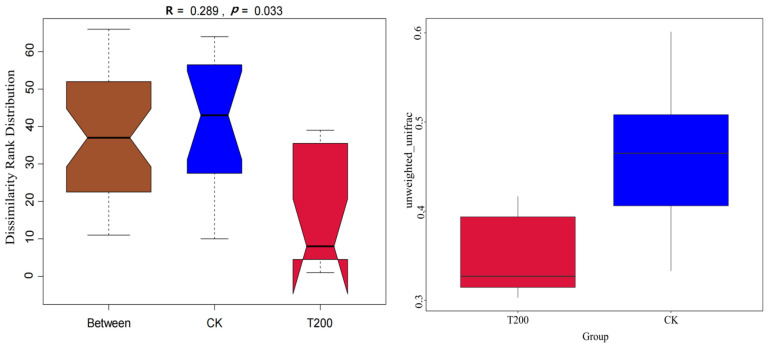
Beta diversity analyses of microbial communities in the rhizosphere soil using anosim and Unweighted_UniFrac. The boundary in the figure represents the upper and lower interquartile intervals (IQR), the horizontal line represents the median, and the upper and lower whiskers in the left figure represent 1.5 times the IQR range beyond the upper and lower interquartile, respectively. “Between” indicates the difference between groups; the thickness represents the sample size. If the gaps in each block diagram are inconsistent with each other, this implies that the median value of each group is different.

**Figure 2 plants-12-01096-f002:**
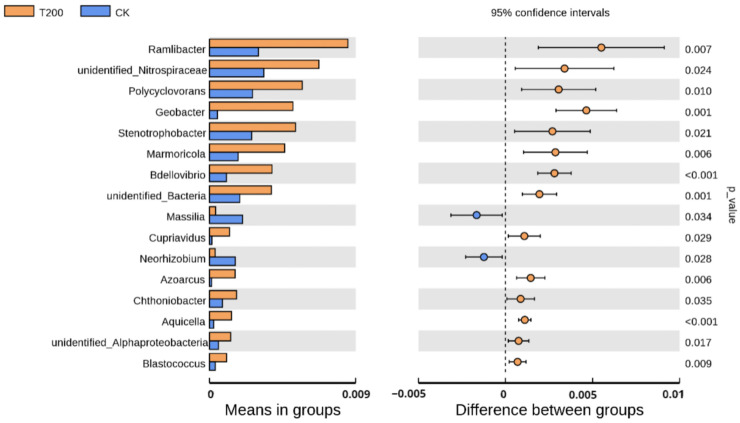
Differences in the relative abundance of bacterial community at genus level among different treatments. The left panel shows the relative abundance of differentially abundant bacterial communities under the two treatments. The right panel shows the *p*-value as the ordinate and the significant difference as the abscissa.

**Figure 3 plants-12-01096-f003:**
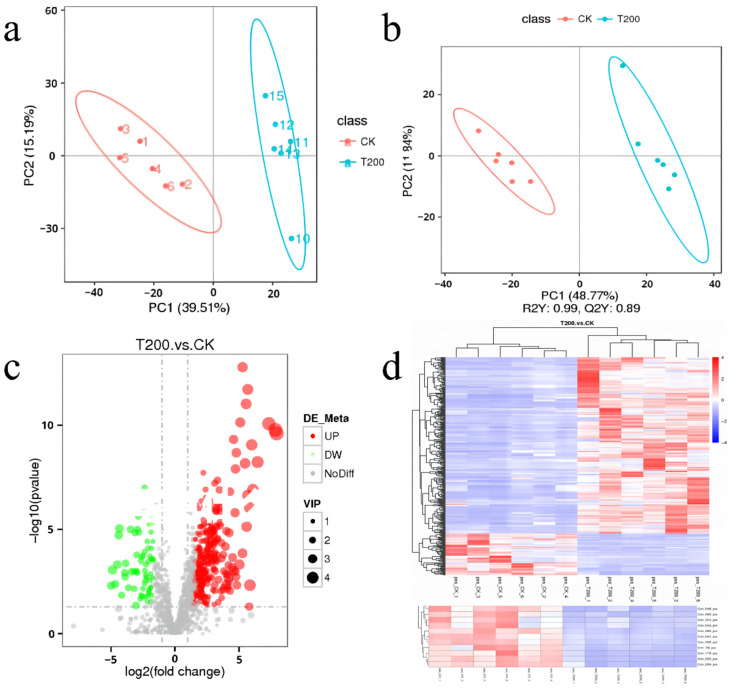
Differential expression analysis of the non-targeted metabolome in roots. (**a**,**b**) depict PCA and PLS-DA analyses, respectively, and each point represents a sample. The distance between two points in the horizontal and vertical coordinates represents the distance similarity of the sample under the influence of the principal components PC1 or PC2. Each point in (**c**) represents the degree of difference in the detected metabolite. The horizontal axis represents the fold change, while the vertical axis represents the FDR-adjusted *p*-value. The color of the dot indicates the expression of the corresponding gene. Red denotes upregulated metabolites, green represents downregulated metabolites, and gray denotes the metabolites that are not differentially expressed. VIP value represents the contribution rate of metabolite difference in different groups. The relative content in (**d**) is shown using different colors. The intense red color corresponds to a higher expression level, while the intense purple color corresponds to a lower expression level. The columns represent the sample, the rows represent the metabolite names, and the cluster tree on the left side of the figure is associated with the differential metabolites.

**Figure 4 plants-12-01096-f004:**
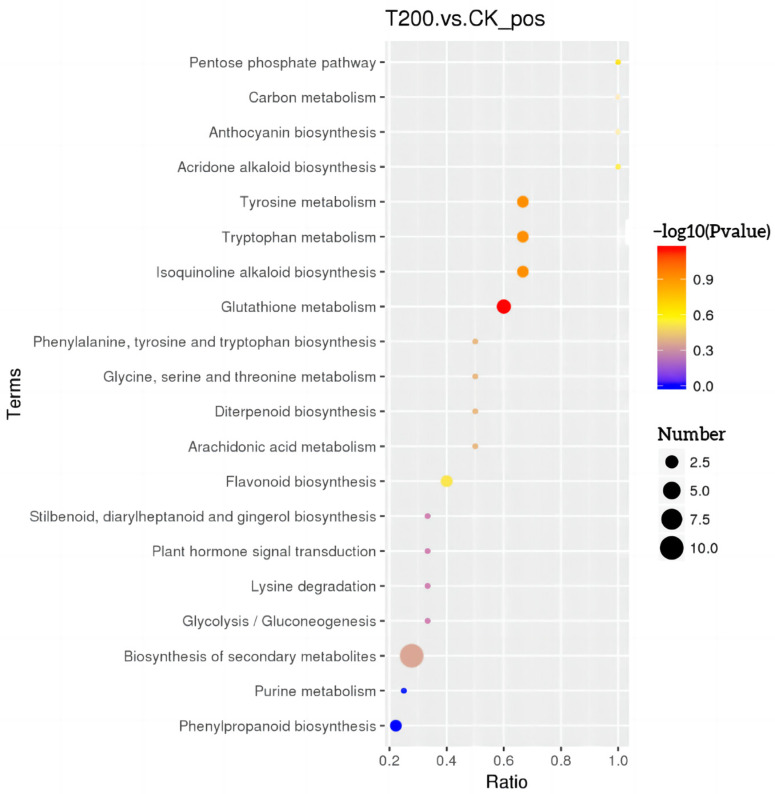
Map representing the KEGG pathway enrichment of non-targeted metabolites in roots. The horizontal axis represents the metabolic pathway ratio of T200 and CK groups, and the vertical axis represents KEGG terms. Different colors represent different *p*-values, ranging from blue at the lowest end of the spectrum (denoting a small *p*-value) to red at the highest end (denoting a large *p*-value). The size of the spot represents the number of microorganisms enriched in a given pathway.

**Figure 5 plants-12-01096-f005:**
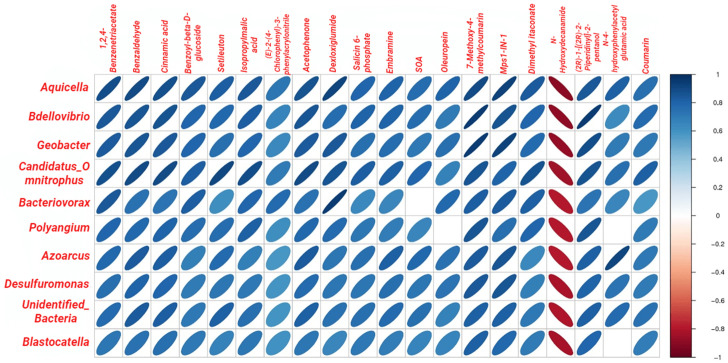
Correlation analysis between bacterial genera in the rhizosphere soil and root metabolites. The thinner the ellipses, the more significant the correlation. The blue-to-red color transition indicates a spectrum ranging from a positive correlation to a negative correlation. Only the metabolites and root metabolites with significant differences between the CK and T200 groups are depicted.

**Table 1 plants-12-01096-t001:** Alpha diversity analysis of bacterial communities in rhizosphere soil.

Items	Treatments		SEM	*p*-Value
	CK	T200		
Observed_species	3349	4031	170	0.015
Shannon	8.951	9.732	0.263	0.485
Simpson	0.987	0.993	0.003	0.572
Chao1	3753.329	4413.943	164.114	0.026
ACE	3823.739	4449.394	158.051	0.041
Goods_coverage	0.989	0.989	0.001	0.737

**Table 2 plants-12-01096-t002:** Differentially expressed metabolites identified in roots using non-targeted metabolomes.

Metabolites	Log_2_FC	Metabolites	Log_2_FC	Metabolites	Log_2_FC
1,2,4-Benzenetriace	5.27	Gefitinib	5.80	Eriocitrin	3.00
Benzaldehyde	5.67	Omapatrilat	−2.44	Tryptophol	2.58
Cinnamic acid	5.55	Gloriosine	2.79	gibberellin A1	4.00
Benzoyl-beta-D-glucoside	5.09	Efinaconazole	1.56	Maculin	2.44
Setileuton	7.31	Pranlukast	3.19	2-O-ETHYL ASCORBIC ACID	2.97
Isopropylmalic acid	7.82	Maltol	3.50	Codeinone	3.07
Dihydroxyphenylalanine	3.22	Cassyfiline	3.22	Levallorphan	2.43
Acetophenone	4.56	Myxochelin C	4.68	Quercetin	−4.89
Dexloxiglumide	5.98	L-Lombricine	−2.86	Drotaverine	−2.82
Salicin 6-phosphate	4.78	MFCD15146035	−1.87	L-serine phosphoethanolamine	2.15
Embramine	6.44	Roquefortine E	2.48	20-hydroxy-leukotriene E4	1.60
SOA	5.46	Andrographolide	2.68	Xanthurenic acid	1.64
Oleuropein	4.72	Chlorogenic acid	−3.00	Stachyflin	2.47
7-Methoxy-4-methylcoumarin	2.21	Lactide	2.57	Naftalofos	1.78
Mps1-IN-1	3.31	Isofraxidin	−2.02	Dinocton 6	2.89
Dimethyl itaconate	3.09	Oxandrolone	2.21	Kuwanon G	3.04
N-Hydroxydecanamide	−2.38	Paramethadione	2.24	Guanidinobutyrate	1.89
TEMPO	1.70	Allantoic acid	5.58	Sennoside C	−4.39
Repirinast	3.31	Istamycin C1	1.92	Azlocillin	2.80
Coumarin	3.47	Nodakenin	2.04	Premithramycin A2′	2.74
8-Bromoguanosine	4.27	Homotrypanothione	2.05	Pevonedistat	−3.46
Golotimod	5.67	Tyramine	2.87	Eriocitrin	3.00
Droxidopa	2.42	S-Bicalutamide	2.91	Taspine	1.59
Oxybutynin	2.05	5-Aminopentanamide	2.83	Bis(glutathionyl)spermine	5.78
Valeramide	2.81	Payzone	−4.34	Bis(glutathionyl)spermine disulfide	2.04
Salidroside	1.82	Glycerophosphoglycerol	−2.93	Levallorphan	2.43
Dasolampanel	3.16	Dibenzylmethane	2.16	7-ACA	−1.87

## Data Availability

Not applicable.

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
