# Peer review of "Effects of Coumarin on Rhizosphere Microbiome and Metabolome of Lolium multiflorum"

_plants, 2023, doi:10.3390/plants12051096_

Round 1

Reviewer 1 Report (Previous Reviewer 2)

The resubmitted manuscript has major improvements in readability. I only have now a few minor suggestions.

Line 12: substitute “between commensals and pathogens and plants” with “between commensals, pathogens and plants”

Line 29: something is missing in the sentence: “a strong was observed”

Lines 121-123: Please provide evidence that the differences in unweighted UniFrac distances are not significant.

Lines 245-247: Please provide a suitable reference.

Line 247: substitute “researches” with “research”

Line 250: substitute “this kind of bacteria” with “Pseudomonadaceae

Line 375: substitute “PH” with “pH”

Line 412: substitute “amputation” with “trimming”

Line 413: please specify the annotation database

Line 420 - 421: “Finally, the data of each sample were normalized, and the data with the least amount of data in the sample were normalized as the standard”.  I believe you mean that you rarified the data to the sample with lowest read number, correct? If so, please reformulate this sentence, so that it is more easily understandable.

Line 422: If you indeed rarified the data to the sample with lowest read number, please substitute “homogeneized” with “rarified”

Author Response

Reviewer 2 Report (New Reviewer)

In this manuscript, the authors have described the Effect of coumarin on rhizobacterial community structure, and root metabolome of Lolium multiflorum Lam. The work looks okay and well presented, as the authors had been able to justify their hypothesis through data. Though, their are some minor concerns that need to be addressed as given below:

Line 17: in abstract - correct ‘rhizosphere’

Line 18: ….significant effect on the abundance of the bacterial community. Probably it should not be community, but either of taxon/phylum/ any taxonomical unit. Or (?)…significant effect on the abundance of the bacteria in rhizospheric microbial community. Please see.

Line 22: metabolomics (not Metabolomics)

Line 56-59: need to be rephrased

Line 82: var should not be italicized

Line 89: avoid using ‘flora’ for microorganisms. Its an old school concept.

Line 92: why some coumarins name start with small other with capital letters?

Heading 2.1: Diversity ‘in’ the Bacterial Community ‘of’ the Rhizosphere Heading 2.2: Whether the abundances of Bacterial “Communities” has been reported here?

Line 140-143 and other places: ‘unidentified’ should not be italicized

Line 197-198, and other places: The correlation has been deduced for bacterial genera, not communities

Discussion is very presumptive. The metabolites had been correlated to functions, with assumptions, without any data or experiment as taken in this research paper. (eg: line 321-328; line 351-355)

Line 375: correct pH

Line 397: KAc??

Line 400: purity of DNA was checked by electrophoresis? Is this correct method?

Line 407: use reactions, instead of rxns

Title: ‘microecology’ is little overstatement it seems, as the authors have only studied the variation in bacterial community structure.  Also, work on root metabolome does not reflect in the title. Instead of annual ryegrass, scientific name may be given.

Author Response

This manuscript is a resubmission of an earlier submission. The following is a list of the peer review reports and author responses from that submission.

Round 1

Reviewer 1 Report

Roots interact with microorganisms by providing resources such as exuded sugars, metabolites, and cells, but during this process they also exude secondary metabolites to which only some microbes are tolerant and thus exert a selective antimicrobial action on the rhizosphere microorganisms. One well characterized group of exuded metabolites with a selective action are the coumarins. In the current manuscript, Yang et al. investigate how coumarin, a metabolite that also has allelopathic activity, when applied in pot experiments, affects root metabolites and rhizosphere bacterial community of annual ryegrass. The results are potentially interesting and after a thourough revision worth for resubmission. Unfortunately. the text is poorly written, many typos are included, the figures of poor quality, the legends not sufficient for understanding the figures, significance of the differences or correlations not reported, and the authors use the wording abundance for amplicon sequence datasets without a qPCR confirmation for differential representation. The results cannot be evaluated in this current form of the manuscript.

ln 143. If you use the word enriched the difference have to be significant. Otherwise the KEGG terms are not enriched.
Again, in the "Correlation analysis between rhizosphere soil bacterial community and root metabolites" it remains unclear if the data are significant or not.
ln 165. The first paragraph should directly relate to the data observed.
ln 248. Provide reference for "Flavonoids can promote the production of glutathione (GSH), superoxide dismutase 248 (SOD), catalase (CAT) and other antioxidant enzymes in plants"
Figure 1. The figure is not understandable by the used legend, and the same applies for other figures and tables of the manuscript. On the left no values are shown. Change the lettering to larger and improve aesthetics.
Some text examples (TO means change to):
-phenylpropane biosynthesis TO phenylpropanoid biosynthesis
-organ substances released from roots TO root exudates
-Therefore, we hypothesis that coumarin can influence the annual ryegrass through mediating the rhizosphere soil microbial diversity and root 94 metabolites. TO Therefore, we hypothesize that coumarin affects rhizosphere microbial diversity and the patterns of root metabolites in annual ryegrass.
-And both anosim analysis and unweighted unifrac distance analysis (Fig.1) showed the great differences of bacterial community between groups CK and T200. TO And, both anosim and unweighted unifrac distance analysis (Fig.1) indicated a difference between the bacterial communities in CK and T200.
-As can be seen from figure 2, TO ... Figure 2,
Change the text completely: Cutadapt (V1.9.1) [55] was used to shear low quality reads, then according to the barcode to get samples data, amputation of barcode and preliminary quality to get raw reads.

Author Response

Reply to comments of reviewers

Dear reviewer,

Thank you for the reviewers' comments concerning our manuscript entitled "Effect of Coumarin on Rhizosphere Microecology of Annual Ryegrass (Lolium multiflorum Lam.)".Those comments are all valuable and very helpful for revising and improving our paper, as well as the important guiding significance to our researches. We have studied comments carefully and have made correction which we hope meet with approval.The reviewers' comments are listed below in gray background text. Our reply is given in normal font, and changes/additions to the manuscript are given in yellow background text.

Responds to the reviewer's comments:

Unfortunately. the text is poorly written, many typos are included, the figures of poor quality, the legends not sufficient for understanding the figures, significance of the differences or correlations not reported.

Response:

Thank the reviewers for their suggestions. We are sorry for the trouble caused by the English writing problem. We have improved the text typography, picture clarity, legend and other related issues proposed by the experts, as well as the difference significance analysis in the figure.

Explanation on “use the wording abundance for amplicon sequence datasets without a qPCR confirmation for differential representation”.

Response:

In this study, because we found some related phenomena in the physiological ecology of lolium multiflorum measured earlier, coumarin has an ecosystem effect on the growth (root length, biomass, MDA content, etc.), we want to see whether there are some differences at the transcriptome level and whether they can correspond to the physiological ecology. In the follow-up study, we will carry out quantitative verification of qPCR and further functional research on these microorganisms.

ln 143. If you use the word enriched the difference have to be significant. Otherwise the KEGG terms are not enriched.

Response:

Thank you very much for discovering this error.We apologize for this grammatical problemand have corrected it based on your suggestions. What we want to express is that metabolites mainly exist in these metabolic pathways. Fig. 5 The analysis on the correlation between rhizosphere soil bacteria and root metabolic substances has been revised because the legend is not clear.

In 165. The first paragraph should directly relate to the data observed.

Response:

Thank you for your suggestion. We have changed the ID name to the corresponding metabolite name.

ln 248. Provide reference for "Flavonoids can promote the production of glutathione (GSH), superoxide dismutase 248 (SOD), catalase (CAT) and other antioxidant enzymes in plants"

Response:

References have been added to the corresponding parts of the paper.

Figure 1. The figure is not understandable by the used legend, and the same applies for other figures and tables of the manuscript. On the left no values are shown. Change the lettering to larger and improve aesthetics.

Response:

Thanks to the experts' suggestions, we have made the coordinate system of the picture clear and added notes so that the meaning expressed in the diagram can be correctly expressed, and improved the clarity of the picture, so that readers can read it clearly

Some text examples

Response:

Thank you very much for discovering this error.We apologize for this grammatical problemand have corrected it based on your suggestions. In addition, we have asked native Englisheditors to polish and modify the manuscript.

Reviewer 2 Report

Yang et al. have submitted a manuscript entitled “Effect of coumarin on rhizosphere microecology of annual ryegrass”.  The authors analyze bacterial communities and metabolites associated with the roots of annual ryegrass and exposed to treatment with coumarin.  Although the motivation is clear, there are many points in need of clarification.

Major comments

Please revise the English language throughout the manuscript. I have pointed several instances on the minor comments section, but there are many more instances that would need correction. In some instances it is not possible to understand the meaning of the sentences.

Nearly all figure legends need to be extended, so that the reader can understand their content. In detail: Figure 1 – Please label the two panels as A and B and explain in figure legend to what test/metric they pertain. What do the lines and boxplots represent (mean, median, 95% confidence intervals)? Please label the y axis of the first panel and add a clear label of the y axis of the second panel (I have no idea what “count” should mean here, I thought these were portraying distances?). Are the asterisks on the second panel the result of significance testing? If so, please add to the legend (and why wasn’t this done for the first panel?). Please also explain why the boxplot referring to T200 in the first panel has two protruding triangles in the lower section; I have not yet seen any boxplots with this shape so I don’t know how to interpret this. Figure 2 – Again, please label the two panels as A and B and explain in figure legend what they are about. What do the x axes of both panels mean? What unit do they have? Also, I believe you mean “Differentially abundant genera between treatments” instead of “Difference of relative abundance in genus level”. Figure 3 – Here the panels have a name, but no explanation in the figure legend. Please give more information regarding the individual panels, especially panels c and d. Figure 4 – I could not understand if this figure contains all KEGG pathways, or just the ones different between treatments. What does ratio (in x axis) means? What does the size of the points represent (it says “Number”, does it mean number of metabolites?)? Instead of a color gradient for the p value, it would be much simpler just to highlight the significantly different ones. Figure 5 – I do not understand why you don’t show the names of metabolites instead of the ID name. To decode this, the reader needs to check table S1. Please substitute the ID name (which had no meaning for the reader) in the figure, by the metabolite name and delete Table S1. Why are these genera and metabolites the only ones displayed, when there were many more? Please clarify in the figure legend. Figure S1 – In the figure legend I think you mean something like this: “Shared bacterial OTUs between CK and T200 groups.” Figure S2 – In the figure legend I think you mean something like this: “Significant differentially abundant phyla between CK and T200 groups”. Figure S3 – The picture refers to a total of 181 compounds, but you mentioned in the text that 1477 compounds were detected. Are these 181, the only ones that could be identified and annotated to a KEGG pathway? If so, make this very clear in the text and in the figure legend.

Please add the scientific name of the study plant in the title, abstract and material and methods.

The almost complete first paragraph of discussion belongs in introduction.

In the discussion, you mention results that are not available in the results section. In lines192-194, you mention the main phyla detected in the samples for the first time (In figure S2 you only listed the ones differentially abundant between treatments). In the same way, in lines 200-202, you mention that Pseudomonadaceae were enriched in T200, but you never showed these results before. In line 273 you mention 1,4-triclorobenzene  for the first time and I cannot find it either in table 2, neither in table S1. Please add all of these results to the result section, at least in the text. In lines 231-233, you mention that the present study confirms that coumarin causes stress in ryegrass, but there are no definite results presented on plant stress; yes, the metabolites and bacterial communities have shifted, but this does not necessarily confirm plant stress. Do you have evidence on plant traits to support this claim?

You explain the higher relative abundance of Proteobacteria in the coumarin treatment with the fact that they are Gram-negative bacteria with fast growth rates, which harbor members with capabilities for nitrogen fixation and the production of polycyclic aromatic hydrocarbons. But is also the case for other bacteria, such as members of Firmicutes, which you did not find enriched. Please comment.

In lines 212-216 you refer to the enrichment of Aquicella in the coumarin treatment and point out that this is a pathogen that negatively influences plant growth, but you provide no references for this. Moreover, the two described species of Aquicella are protozoan pathogens and were isolated from water samples. Also, after a little research I couldn’t find reports on Aquicella influence on plant performance and growth. Please provide suitable references for these claims.

In lines 235-239 you point out the importance of tryptophan as a precursor of plant growth hormones, but I fail to grasp the connection with the results. I do not see evidence in the results that substantiates this amino acid being significantly enriched in the coumarin treatment samples. Please provide these results, and link them to what you wrote in these lines.

In lines 250-253 you list the different metabolic pathways that were affected by treatment with coumarin and then try to explain why this could be the case. However, most of these (as far as I can tell from figure 4) were not statistically different between treatments (please feel free to correct me if I misinterpreted the figure). Please comment and adjust this section accordingly.

The conclusions should come after the section for discussion and not after the material and methods.

Minor comments

Line 12:  replace “bacterial species” with “bacterial communities” .

Line 15: please explain what you mean with endemic. Endemic to what?

Line 21: please rephrase the sentence as to not include “and so on”.

Lines 30-33: You start these three sentences in the same way. One could condense the information like the following: “Coumarins are important plant secondary metabolites and the main allelochemicals of many plant species [1], such as species of Umbelliferae, Rutaceae, Leguminosae, and Compositae [2,3].”

Lines 33-34: Please provide references.

Line 52: “they are also the first parts to sense soil stress.” What do you mean by soil stress? This is not common terminology. Plus, stress is something imputed to living organisms. I think what you mean is something like: “they are also the first to sense changes in edaphic properties”.

Line 53: please substitute “their” with “coumarin”.

Line 54: one cannot inhibit root length, but root growth. Please rectify.

Line 57: “had promoting effect of root” in what sense (root size, root health,…)?

Line 63: “induce the accumulation of coumarin compounds in vitro” accumulation where? Please clarify.

Lines 78-81: Please rephrase this sentence. Specifically, I could not understand the last half of the sentence.

Lines 92-93: Please clarify what you mean with “the major portion of microbial biodiversity is created by secondary metabolites”.

Line 101: Please substitute “identical” by “shared”

Line 102: Please substitute “observed_species” by “richness”

Line 105: What do you mean by “great differences of bacterial communities between groups CK and T200.”? In Fig 1, only for the first panel do you show distances between the two groups.

Lines 125-126: Please add the information on how many metabolites could be annotated.

Line 147: You should not start a sentence with “and”. Please rephrase. Please also make clearer that significant differences were only found for Glutatione metabolism.

Line 156: Please substitute “existed” by “had a”. In this section please clarify why you onla show these relationships, when many more would exist (more bacterial genera, more metabolites). Are these only the significant correlations?

Line 207: Please substitute “deficiency” by “decrease”

Line 216: I have not heard the terminology “bloom” being applied to terrestrial ecosystems. Please re-check.

Line 219: Please substitute “had a significant influence on” by “were involved in”

Line 227: Please substitute “can oxidative pentose phosphate pathway” by “can stimulate oxidative pentose phosphate pathway”

Line 261: You mention a significant influence of coumarin on several pathways. As I mentioned before, it doesn’t seem that these were really significantly enriched in the coumarin treated samples. Please clarify.

Line 263-264: Here I think you mean that some of the differentially abundant metabolites were chlorogenic acid and coumarin, since there were more than these, from the results of table 2.

Lines 288-290: Please provide more information regarding the soil properties, such as pH.

Lines 293-295: Please rephrase. Also what do you mean with coumarin and soil quality?

Line 311: Please substitute “refrigerator” with “freezer”.

Line 314: Either detail the extraction method, or provide a suitable reference.

Line 334: I do not understand what you mean by “homogeneized”. Do you mean rarified to minimum sequence number? Please clarify.

Line 338: Why did you choose unweighted UniFrac distances and not weighted? Also, please substitute “unifrac” by “UniFrac” throughout the manuscript.

Line 345: Please substitute “A some of supernatant was diluted” by “The supernatant was diluted”

Round 2

Reviewer 1 Report

The manuscript has imporved from the last version but is not yet at the standard of Plants.

The language has to be improved. The authors should send the manuscript for language improvement to an expert editing firm / a colleague who is a native speaker.

Figures are better, but still not optimal.

Figure 1. Figure can be smaller and the text should be placed in an angle and letter sizes considerably increased.
Change lettering in Figure 5. Now of poor resolution and differs between the rows.
Mark in Figure 5 significant correlations (P<0.05) by an asterisk

Reviewer 2 Report

The authors have revised the manuscript based on the recommendations by the reviewers, but there are still some points that would benefit from further revision.

Although much improved, I still have some difficulties with understanding the figures and figure legends. Legend of figure 1: Substitute “upper and lower tentacles” by “upper and lower whiskers”. What do you mean by “Others in the group”? Legend of figure 2: Substitute “shows the relative abundance of microbial communities” by “shows the relative abundance of differentially abundant bacterial genera.” Legend of figure 3: Please condense a bit the description of panel c. Also, please explain what you mean by “the greater the difference multiple” and “vip value”. Legend of figure 4: I am not sure I understood what you mean by “metabolic pathway ratio”. For example, does a ratio of 0.2 represent that you have 0.2x more of these metabolites in one group versus the other? This is not so clear to me. Legend of figure 5:  By “The greater the proportion of ellipses, the more significant the correlation is”, do you mean the thickness of the ellipses? In this case I would write “The thinner the ellipses, the more significant the correlation”.

You added Figure S3, in order to show the differences in relative abundances between groups. However, from this figure it is only really possible to see that they are significantly different, but not the magnitude of the difference. In particular, at some stage in the discussion you focus on the family Pseudomonadaceae, since this had significant differences between groups. However, when looking at the top panel of Figure S3, the difference in relative abundance appear to be very small. Please add to the text the exact values for each group, so that the reader can know exactly what the difference is.

Regarding the genus Aquicella, you referred now to a paper which shows the presence of one Aquicella OTU only in the rhizosphere of asymptomatic trees, but you used it to substantiate the claim that this has a negative influence in plant growth and development. If this genus is found only in asymptomatic plants, this is indicative of a protective effect and not the contrary. Please revise.

Also, you mention in lines 256-266 that “while this kind of microorganisms has an antagonistic effect on Aquicell, leading to the mass propagation of Aquicella.” Antagonism means opposition, and therefore I think you mean the opposite of what you say. On a smaller note, I must reiterate that research has only showed that these are protistan pathogens. No direct link has been established between this genus and plant disease, so please refer in the text that these are protistan pathogens.

Please ensure that all validly published taxonomic names are italicized.
